# Reply to Henriksen, S.; Rinaldo, C.H. Should SVGp12 Be Used for JC Polyomavirus Studies? Comment on “Prezioso et al. COS-7 and SVGp12 Cellular Models to Study JCPyV Replication and MicroRNA Expression after Infection with Archetypal and Rearranged-NCCR Viral Strains. *Viruses* 2022, *14*, 2070”

**DOI:** 10.3390/v15010093

**Published:** 2022-12-29

**Authors:** Carla Prezioso, Ugo Moens, Valeria Pietropaolo

**Affiliations:** 1IRCCS San Raffaele Roma, Microbiology of Chronic Neuro-Degenerative Pathologies, 00163 Rome, Italy; 2Department of Public Health and Infectious Diseases, Sapienza University of Rome, 00185 Rome, Italy; 3Department of Medical Biology, Faculty of Health Sciences, University of Tromsø—The Arctic University of Norway, 9037 Tromsø, Norway

**Keywords:** BKPyV, JCPyV, SVGp12, infection, replication, contamination

## Abstract

In relation to the comment by Henriksen and Rinaldo, the authors intend to emphasize that before every experiment with SVGp12 cells they routinely test the cells for the absence of BKPyV contamination. The scientists can state that the SVGp12 cells used in their laboratory were not infected by BKPyV and that their results were also validated on the COS-7 cell line, which is permissive for JCPyV infection. Therefore, the overall findings of the study and its conclusions remain authentic. The authors recommend the necessity of carefully testing SVGp12 cells for BKPyV infection before use or, alternatively, in case of a first purchase; moreover, it is possible to choose different cell lines to avoid running into this unpleasant situation.

We thank Stian Henriksen and Christine Hanssen Rinaldo for the opportunity to provide a further point of view on the infection of BK polyomavirus (BKPyV) in the commercially immortalized human fetal glial cell line SVGp12, available from American Type Culture Collection (ATCC) [1].

“The impact of the BKPyV contamination on SVG research is presently difficult to estimate since many studies utilizing SVG- derived cell lines lack a clear specification of the source of the cells. This leads us to reemphasize the implications of our recent paper: researchers who have been using or are using SVG-derived cell lines should test their cells for the presence of BKPyV, and reviewers of such work should demand this testing in case this has not been performed”. This sentence, published by Rinaldo et al. in 2014 in Journal of Virology [2], clearly urges researchers who have been using or are using SVG-derived cell lines to test their cells for the presence of BKPyV.

The laboratory of the authors where the study entitled “COS-7 and SVGp12 Cellular Models to Study JCPyV Replication and MicroRNA Expression after Infection with Archetypal and Rearranged-NCCR Viral Strains” was performed [3] has been carrying out polyomavirus research since 1990 and is one of the most important Italian laboratories for that very research, as documented by the authors’ publications on the matter. Since the expertise of this group is polyomaviruses, the researchers were perfectly aware of the paper by Henriksen et al., published in 2014 [4], of a letter to the editor by Ferenczy and Major published in the same year [5], and of the author reply by Rinaldo and colleagues [2].

On the basis of what has been published and suggested by Rinaldo et al. in 2014 [2], the SVGp12 cell line, owned by the laboratory, was tested in order to exclude the possibility of the BKPyV contamination. Specifically, before every study involving SVGp12 cells and JCPyV and also as an initial characterization of the cells, each vial containing this type of cell was thawed and cultured in Eagle’s minimal essential medium (EMEM), supplemented with penicillin, streptomycin, and fetal bovine serum (FBS), incubated at 37 °C in the presence of 5% CO_2_, and propagated. At the established time points (3, 5, and 7 days post thaw), cells were detached and analyzed for the presence of BKPyV DNA.

Specifically, total DNA was extracted from 1 × 10^6^ cells using a QIAamp^®^ DNA Mini Kit (Applied Biosystems, Waltham, MA, USA). Supernatants (SPNTs) from cells were initially subjected to cycles of freezing and thawing, then centrifuged. The resulting clarified supernatants were used directly in molecular biology assays. Extracted DNA from the cells and SPNTs were analyzed using quantitative real-time PCR (qPCR) targeting a conserved region of the BKPyV VP1 gene. qPCR was performed using a 7300 Real Time PCR System (Applied Biosystems, Waltham, MA, USA). Primers including BKVPf (5′-AGTGGATGGGCAGCCTATGTA-3′, nt 2511–2531), BKVPr (5′-TCATATCTGGGTCCCCTGGA-3′, nt 2605–2586), and Taqman MGB probe BKVPp (5′FAM-AGGTAGAAGAGGTTAGGGTGTTTGATGGCACAG-3′MGB, nt 2578–2546) were used for the amplification and detection of the target sequence. The reaction was performed in a final volume of 25 μL containing 1× Taqman Universal PCR Master Mix (Applied Biosystems, Waltham, MA, USA), 0.4 μM of primer BKVPf, 0.9 μM of primer BKVPr, 0.2 μM of BKVPp, and 5 μL of extracted nucleic acid. Thermal cycling was carried out according to the following steps: an initial denaturation at 95 °C for 10 min followed by 40 cycles at 95 °C for 15 s and at 60 °C for 1 min, after which the fluorescence was read. Each sample was analyzed in triplicate, and each run contained a negative control with the reaction mixture and without a DNA template. A standard curve for BKPyV quantification was constructed using serial dilutions of a plasmid containing the whole BKPyV genome (range: 10^2^ to 10^6^ copies). The detection limit for this assay was determined to be 5 copies/reaction [6].

Using these experimental conditions, we failed to detect BKPyV sequences in our SVGp12 cells. Because of negative results of the BKPyV DNA detection, the researchers deemed the SVGp12 cell line in their possession to be suitable for experiments involving JCPyV.

In relation to the comment by Henriksen and Rinaldo [1] on Prezioso et al.’s “COS-7 and SVGp12 Cellular Models to Study JCPyV Replication and MicroRNA Expression after Infection with Archetypal and Rearranged-NCCR Viral Strains” [3], the authors intend to answer the following questions.

## 1. Could the Persistent BKPyV Infection Have Been Detected with the Methodologies Used in Prezioso et al.’s Study?

This question is easy to answer since, as described above, before any experiment involving SVGp12 cells and JCPyV infection the cells were tested for the presence of BKPyV. Because of the apparent lack of BKPyV in our SVGp12 cells, it is therefore possible to assume that all the methodologies applied in the paper by Prezioso et al. were specifically for JC viral investigation and for the detection of microRNAs specific to the virus.

## 2. In What Way May the Presence of BKPyV Have Affected Their Results?

Since we certainly can state that our laboratory has not used cells infected by BKPyV and that our results were also validated on the COS-7 cell line, which is permissive for JCPyV infection, the overall findings of Prezioso et al.’s study “COS-7 and SVGp12 Cellular Models to Study JCPyV Replication and MicroRNA Expression after Infection with Archetypal and Rearranged-NCCR Viral Strains” [3] and its conclusions remain authentic.

As further evidence supporting the absence of BKPyV infection in the SVGp12 cells in the study by Prezioso et al. [3], the authors studied the expression of JCPyV miR-J1-5p in infected cells and purified exosomes and also the effects of pre-treatment with exosomes before infection. It has been shown that JCPyV encodes a pre-miRNA that is processed into two unique miRNAs, JCPyV-specific miR-J1-5p and miR-J1-3p, during the late phase of infection [7]. To further confirm that our SVGp12 cells were not infected by BKPyV, we firstly focused our study on both miR-J1-5p and miR-J1-3p expression. MiR-J1-3p miRNA has complete sequence homology to BKPyV miR-B1-3p. MiR-J1-3p was never detected, causing us to focus our attention only on the miR-J1-5p expression that is more specifically representative of JCPyV. The absence of 3p miRNA may also indicate that BKPyV was not present in the SVGp12 cells, although we cannot conclude this with certainty because, at least for JCPyV, miR-J1-5p levels seemed to be dominant over those of miR-J1-3p [8]. We did not test our SVG 12p cells for the presence of miR-B1-5p because our qPCR with BKPyV VP1 specific primers was negative.

Furthermore, regarding the ability of BKPyV to influence JCPyV in infecting SVGp12 cells, it must be underlined that BKPyV infection is not needed for JCPyV to infect and replicate in SVGp12 cells, which has also reported in previous studies by Henriksen and colleagues [4]. For JCPyV replication, the most important characteristic of these cells is SV40 LTag expression.

## 3. Should SVGp12 Be Used for JCPyV Studies?

The authors of a study from 2016 claimed not to have found BKPyV in their SVGp12 cells [9]. Moreover, the SVGp12 cells in our possession were purchased from ATCC in the first months of 2016.

In conclusion, we underline the authenticity of our findings, although we do not argue the indisputable evidence of productive BKPyV infection in at least five ATCC vials of SVGp12 cells, supported by Henriksen et al. [4].

We join the suggestion to carefully test SVGp12 cells for BKPyV infection before use or, alternatively, in case of a first purchase; moreover, it is possible to choose different cell lines to avoid running into this unpleasant situation.

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
