# Peer review of "Reply to Henriksen, S.; Rinaldo, C.H. Should SVGp12 Be Used for JC Polyomavirus Studies? Comment on “Prezioso et al. COS-7 and SVGp12 Cellular Models to Study JCPyV Replication and MicroRNA Expression after Infection with Archetypal and Rearranged-NCCR Viral Strains. Viruses 2022, 14, 2070”"

_viruses, 2022, doi:10.3390/v15010093_

Round 1

Reviewer 1 Report

In this manuscript, the authors have replied to the comments from Drs. Henriksen and Rinaldo regarding their published paper. 

The commenters noted that the SVG p12 cells used in this study may have been contaminated by BK polyomavirus (BKV) and raised these concerns with the authors. 

In this reply, the authors have stated that they recognize the possibility of BKV contamination of SVG p12 cells and that these cell lines have been confirmed by highly sensitive testing to be free of BKV infection. They also provided clear and evidence-based answers to questions asked by the commenter. 

This reviewer believes that the discussion among the commenters and authors on this paper will provide more valuable knowledge about SVG p12 cells that will be useful in studying JC polyomavirus, which have a narrow host cell range.

Author Response

This reviewer does not request any review to the paper.

Reviewer 2 Report

Dear authors,

Please find below some modifications I suggest before the publication of your reply to “Should SVG p12 be used for JC polyomavirus studies? Comment on COS-7 and SVG p12 Cellular Models to Study JCPyV Replication and MicroRNA Expression after Infection with Archetypal and Rearranged-NCCR Viral Strains" by Henriksen et al..

Minor points :

L33 : « clearly urges » or « clearly urged » instead of « clearly urging » ?

L49 : « 37°C » instead of « 37oC »

L50 : « CO2 »instead of « CO2 »

L88 : « certainly » instead of « certainty »

L92 : « remain » instead of « remains »

L102-104 : This sentence is unclear.
Probably
« The absence of 3p miRNA may also indicate the BKPyV was not present in the SVGp12 cells, although we can not conclude this with certainty because, at least for JCPyV, miR-J1-5p levels seems to be dominant over miR-J1-3p [8]. »
instead of
« The absence of 3p miRNA may also indicate the BKPyV was not present in the SVGp12 cells, although we conclude this with certainty because, at least for JCPyV, miR-J1-5p levels seems to be dominant over miR-J1-3p [8]. »

L105 : « miR-B1-5p » instead of « mi-B1-5p »

L120 : « it is also possible to choose » instead of « it is also possible choose »

Author Response

Author's Reply to the Review Report (Reviewer 2)

Comments and Suggestions for Authors

Dear authors,

Please find below some modifications I suggest before the publication of your reply to “Should SVG p12 be used for JC polyomavirus studies? Comment on COS-7 and SVG p12 Cellular Models to Study JCPyV Replication and MicroRNA Expression after Infection with Archetypal and Rearranged-NCCR Viral Strains" by Henriksen et al.

Minor points :

L33 : « clearly urges » or « clearly urged » instead of « clearly urging » ?

Thank you for the suggestion. We replaced “clearly urging” with “clearly urges”.

L49 : « 37°C » instead of « 37oC »

Thank you for the suggestion. We replaced “37oC” with “37°C”.

L50 : « CO2 »instead of « CO2 »

Thank you for the suggestion. We replaced “CO2” with “CO2”.

L88 : « certainly » instead of « certainty »

Thank you for the suggestion. We replaced “certainty” with certainly.

L92 : « remain » instead of « remains »

Thank you for the suggestion. We replaced “remains” with “remain”.

L102-104 : This sentence is unclear.

Probably

« The absence of 3p miRNA may also indicate the BKPyV was not present in the SVGp12 cells, although we can not conclude this with certainty because, at least for JCPyV, miR-J1-5p levels seems to be dominant over miR-J1-3p [8]. »

instead of

« The absence of 3p miRNA may also indicate the BKPyV was not present in the SVGp12 cells, although we conclude this with certainty because, at least for JCPyV, miR-J1-5p levels seems to be dominant over miR-J1-3p [8]. »

Thank you for the suggestion. We replaced “The absence of 3p miRNA may also indicate the BKPyV was not present in the SVGp12 cells, although we conclude this with certainty because, at least for JCPyV, miR-J1-5p levels seems to be dominant over miR-J1-3p [8]”

with:

“The absence of 3p miRNA may also indicate the BKPyV was not present in the SVGp12 cells, although we cannot conclude this with certainty because, at least for JCPyV, miR-J1-5p levels seems to be dominant over miR-J1-3p [8]”.

L105 : « miR-B1-5p » instead of « mi-B1-5p »

Thank you for the suggestion. We replaced “mi-B1-5p” with “miR-B1-5p”.

L120 : « it is also possible to choose » instead of « it is also possible choose »

Thank you for the suggestion. We replaced “it is also possible choose” with “it is also possible to choose”.